# Ni-catalyzed enantioconvergent deoxygenative reductive cross-coupling of unactivated alkyl alcohols and aryl bromides

Li-Li Zhang [1,4], Yu-Zhong Gao [2,4], Sheng-Han Cai [1], Hui Yu [1], Shou-Jie Shen[2], Qian Ping[3] & Ze-Peng Yang [1] ✉

Transition metal-catalyzed enantioconvergent cross-coupling of an alkyl precursor presents a promising method for producing enantioenriched C(sp$^3$) molecules. Because alkyl alcohol is a ubiquitous and abundant family of feedstock in nature, the direct reductive coupling of alkyl alcohol and aryl halide enables efficient access to valuable compounds. Although several strategies have been developed to overcome the high bond dissociation energy of the C−O bond, the asymmetric pattern remains unknown. In this report, we describe the realization of an enantioconvergent deoxygenative reductive cross-coupling of unactivated alkyl alcohol (β-hydroxy ketone) and aryl bromide in the presence of an NHC activating agent. The approach can accommodate substituents of various sizes and functional groups, and its synthetic potency is demonstrated through a gram scale reaction and derivatizations into other compound families. Finally, we apply our convergent method to the efficient asymmetric synthesis of four β-aryl ketones that are natural products or bioactive compounds.

The saturation degree and the presence of chiral centers are two factors that correlate to the successful transition of a compound from discovery, to clinical testing, and ultimately into a drug[1]. Transition metal-catalyzed enantioconvergent cross-coupling of an alkyl precursor presents a promising method for producing these molecules[2–7]. This approach has been shown to be highly effective in forging C(sp$^3$)−C(sp$^2$) bond, particularly when using nickel as the catalyst[8–18]. Traditional coupling utilizes alkyl halide and organometallic reagent to form a new C−C bond (Fig. 1a)[19–22]. The application of reductive cross-coupling, led by Weix[23–29], Reisman[30–40], and others[41–64], has proven beneficial in circumventing the utilization of organometallic reagents that are vulnerable to air and moisture, and in shortening the synthesis with fewer steps. On the other hand, alkyl halide can be produced through the Appel reaction using alkyl alcohol. Alkyl alcohol is an abundant alkyl source in nature and would be a desirable choice for

C(sp$^3$) coupling. However, the direct cross-coupling of alkyl alcohols remains an underdeveloped field, especially when creating asymmetric patterns[65–69].

Due to the high bond dissociation energy of the C−O bond and the low leaving ability of the OH$^-$ group[70], the direct reductive cross-coupling of alkyl alcohols with aryl halides is elusive. In this vein, many types of alcohol derivatives, such as alkyl acetates[71,72], tosylates[73–75], xanthate esters[76], mesylates[77], pivalates[78,79], oxalates[80–82], methyl ethers[83], chloroformates[84], and others[85–87], have been extensively explored in reductive cross-coupling reactions. However, most of these methods are limited to activated alkyl alcohol derivatives, and the pre-activation requires additional steps. In 2018, Ukaji and coworkers offered an appealing solution by employing a Ti-mediated direct reductive cross-coupling, which only works with primary benzyl alcohols[88–98]. An alternative strategy that may be used in certain cases

[1]School of Chemical Science and Engineering, Tongji University, Shanghai 200092, People's Republic of China. [2]Key Laboratory of Magnetic Molecules, Magnetic Information Materials Ministry of Education, The School of Chemical and Material Science, Shanxi Normal University, Taiyuan 030031, People's Republic of China. [3]State Key Laboratory of Pollution Control and Resource Reuse, College of Environmental Science and Engineering, Tongji University, Shanghai 200092, People's Republic of China. [4]These authors contributed equally: Li-Li Zhang, Yu-Zhong Gao. ✉e-mail: zpyang@tongji.edu.cn

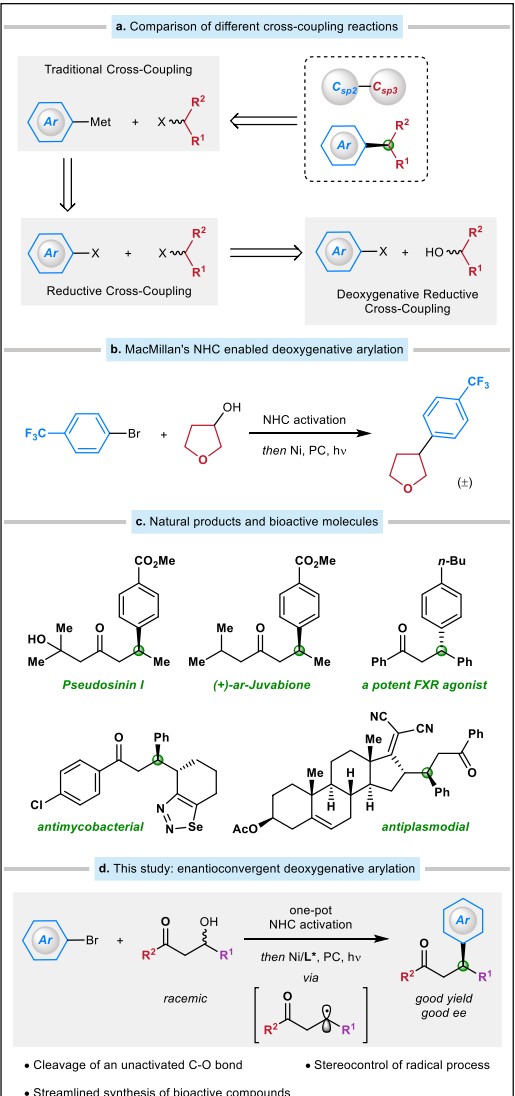

**Fig. 1 | Background of this study. a** Comparison of different cross-coupling reactions. **b** A breakthrough from the MacMillan group. **c** Examples of natural products and bioactive molecules. **d** This study: Ni-catalyzed enantioconvergent deoxygenative reductive cross-coupling of unactivated alkyl alcohols and aryl bromides. Met metal complex, NHC N-aryl benzoxazolium salt, PC photocatalyst.

is the one-pot process involving the in situ activation of alkyl alcohols and their subsequent reductive cross-coupling. In this context, pioneering studies from Li[99,100], Gong[101], Weix[102], Shu[103–106], and others[107–113] have demonstrated the power of this approach owing to the ubiquity of the two building blocks: free alcohols and aryl bromides. A breakthrough was disclosed by the MacMillan group, who realized an NHC (N-heterocyclic carbene) enabled deoxygenative arylation in 2021 (Fig. 1b)[114–119]. This robust method features mild conditions and simple operations, and targets a broad spectrum of primary, secondary, and tertiary alcohols.

Despite these significant advances, an enantioconvergent deoxygenative reductive cross-coupling of an alkyl alcohol, especially an unactivated alkyl alcohol, still needs to be addressed. Managing the stereoselective nature of free radical reactions remains a formidable task because of the high reactivity of radical species. β-Hydroxy ketones are a readily available building block that can be obtained by one-step aldol condensation and contain an unactivated alkyl alcohol group. Furthermore, the carbonyl group is an essential functional group in organic chemistry due to its versatility in forming a variety of

structures and its prevalence in numerous biologically relevant compounds. We speculate that MacMillan's robust NHC system would serve as an excellent foundation for achieving the enantioconvergent deoxygenative reductive cross-coupling of β-hydroxy ketones, providing ready access to a wide variety of β-aryl ketones that are a common subunit found in many natural products and bioactive molecules (Fig. 1c). Herein we describe the realization of this objective by using a chiral nickel/pyridyloxazoline catalyst (Fig. 1d).

## Results

### Reaction optimization

In an initial study, we examined the coupling of racemic 3-hydroxy-1-phenyl-1-heptanone with methyl 4-bromobenzoate (Table 1). We chose an N-aryl benzoxazolium salt (NHC) as the activator to convert the alkyl alcohol to an NHC-alcohol adduct in situ. It's worth noting that this benzoxazolium salt can be easily prepared in the lab on a hundred-gram scale. After an extensive evaluation of all reaction parameters, we determined that NiBr$_2$·DME and chiral pyridyloxazoline ligand **L1** can accomplish the desired enantioconvergent deoxygenative reductive cross-coupling in good yield and ee (83% yield, 92% ee; entry 1).

In the absence of NiBr$_2$·DME, photocatalyst, quinuclidine, light, or ligand **L1**, essentially no or only a small amount of product is observed (racemic; entries 2 and 3). The presence of 4-methylpyridine is crucial in the deoxygenative reductive coupling process. The absence of this additive results in a product with much lower efficiency and selectivity (entry 4). A variety of other chiral ligands are less effective than ligand **L1** (entries 5–9). Moreover, the mixed solvent proves superior to a single solvent (entries 10 and 11). Employing any other base or additive results in a subpar outcome (entries 12 and 13). If the coupling is conducted with less catalyst, for a shorter time, or at an elevated temperature (r.t.), then a lower yield and/or ee are obtained (entries 14-16). The reaction proceeds relatively smoothly in the presence of a small amount of air (entry 17), whereas a reaction run with water leads to a diminished yield and ee (entry 18). Under these conditions, the corresponding ester, amide, and phosphonate are not suitable coupling partners (entry 19). Additionally, substrates like 2-butanol and others that have a functional group at the β-position (-Ph, -OBz, -NHCbz, -NHBz, -NPh$_2$, etc.), were found to provide much lower yield and/or ee (for a broad exploration of other potential substrates, see Supplementary Fig. 6 and Supplementary Fig. 7), highlighting the critical role of the ketone moiety as a potential directing group in the cross-coupling reaction[120]. It is also noteworthy that the MacMillan group employed a similar pyridyloxazoline ligand to successfully couple alkyl alcohols and alkyl bromides in an achiral/racemic manner, which offers further support to the notion that the ketone moiety used in this study plays a significant role in enantiocontrol[118].

### Substrate scope

With the optimized reaction conditions in hand, we sought to examine the generality of substrate scope for both coupling partners. This straightforward method for the catalytic enantioconvergent synthesis of arylated products is compatible with an array of substituents at the β-position (R[1]; Fig. 2a) of the ketones, providing a range of products with good yields and high ee. For example, the alkyl substituent at the β-position can vary in size from methyl to neopentyl to isopropyl, and consistently good yields and ee are observed (products **1**–**7**). A variety of functional groups can be present, including silyl ether, ether, ester, unactivated primary alkyl fluoride/chloride, terminal olefin, and Boc-protected amine (products **8**–**18**). In reactions involving alcohols with a stereocenter, the catalyst determines the stereochemistry outcome, rather than the substrate (products **19** and **20**). Notably, the presence of an aryl group at the β-position in place of an alkyl group results in a comparable outcome as well (products **21**–**23**).

With regard to the groups attached to the carbonyl of the ketones (R[2]; Fig. 2b), many aryls prove to be appropriate, including several

**Table 1 | Effect of reaction parameters on the enantioconvergent deoxygenative reductive cross-coupling reaction**

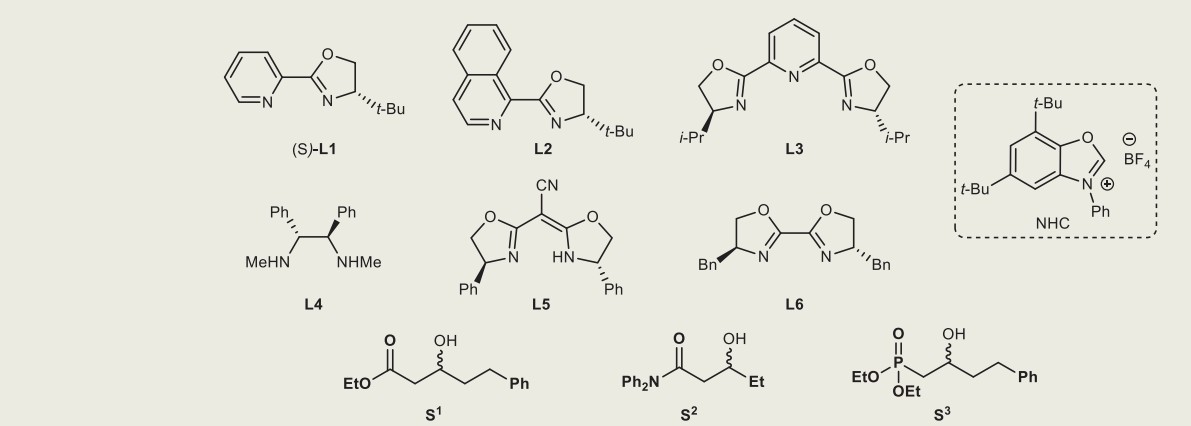

| entry | variation from the "**standard conditions**" | yield (%)[a] | ee (%)[b] |
|---|---|---|---|
| 1 | None | 83 | 92 |
| 2 | No Ni, PC, Quinuclidine, or light | 0 | – |
| 3 | No (S)-**L1** | 13 | 0 |
| 4 | No 4-Methylpyridine | 23 | 25 |
| 5 | **L2**, instead of (S)-**L1** | 18 | 60 |
| 6 | **L3**, instead of (S)-**L1** | 12 | –34 |
| 7 | **L4**, instead of (S)-**L1** | 15 | 25 |
| 8 | **L5**, instead of (S)-**L1** | 0 | – |
| 9 | **L6**, instead of (S)-**L1** | 30 | 31 |
| 10 | Pure MTBE | 62 | 82 |
| 11 | Pure *i*-PrOH | 2 | 87 |
| 12 | NaOAc, instead of Quinuclidine | 13 | 84 |
| 13 | Pyridine, instead of 4-Methylpyridine | 70 | 88 |
| 14 | 2.5 mol% NiBr$_2$•DME, 3.0 mol% (S)-**L1** | 39 | 92 |
| 15 | 9 h, instead of 18 h | 67 | 92 |
| 16 | r.t., instead of 10 °C | 23 | 88 |
| 17 | 1.0 mL air added (4 mL reaction vial) | 70 | 89 |
| 18 | 0.1 equiv H$_2$O added | 32 | 80 |
| 19 | **S$^1$**, **S$^2$**, or **S$^3$**, instead of β-hydroxy ketone | <1 | – |

*MTBE* methyl *tert*-butyl ether, *DME* 1,2-dimethoxyethane, *PC* (Ir[dF(CF$_3$)ppy]$_2$(dtbpy))PF$_6$, *LED* light emitting diode. The green circle signifies an enantioenriched compound.
[a]Determined through GC analysis.
[b]Determined through HPLC analysis.

heteroaryls such as furan, thiophene, and benzothiophene (products **24**–**36**). Furthermore, not only aryl ketones but also alkyl ketones illustrate superior reactivity and selectivity, and the alkyl size can vary in size from methyl to *tert*-butyl (products **37**–**41**).

We next evaluated the scope of aryl bromides (Ar; Fig. 2c). Under similar conditions, the chiral nickel catalyst couples 1.0 equivalent of racemic β-hydroxy ketone to provide the substitution product with good enantioselectivity and yield (for example, product **44**, 70% yield, 93% ee). The observed values of the enantiomeric excess and

yield provide evidence that the coupling reaction works as an enantioconvergent process. In this context, the catalyst efficiently converts both enantiomers of the racemic alkyl alcohol substrate into a specific stereoisomer of the desired product. This protocol can efficiently incorporate aryl bromides containing either electron-rich or electron-deficient substituents, complementing previously established Ni-catalyzed reductive cross-couplings that are typically limited to the electron-deficient aryl halides. Many functional groups, including ester, fluoride/chloride, trifluoromethyl, Bpin, nitrile, and

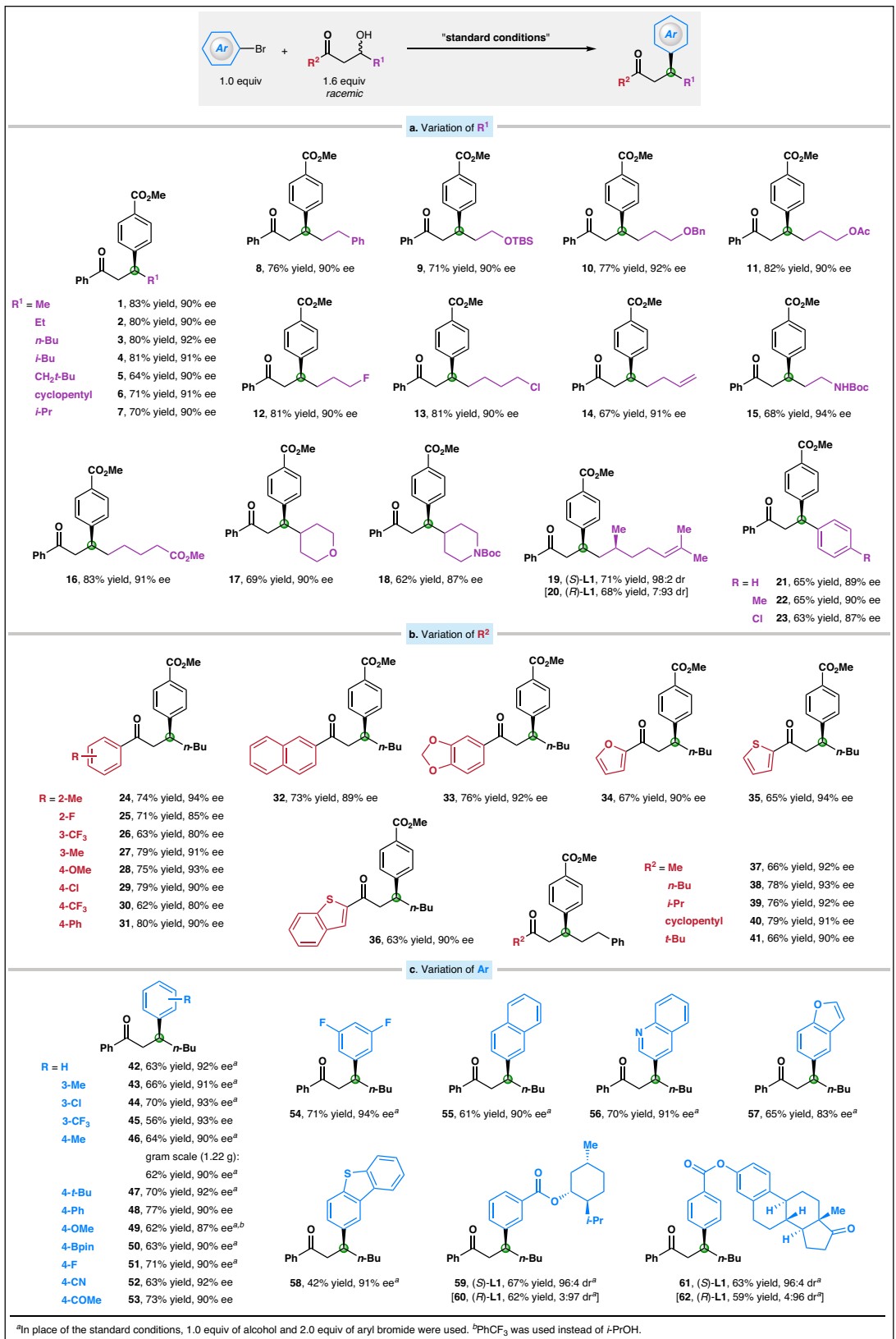

**Fig. 2 | Scope of the catalytic enantioconvergent deoxygenative reductive cross-coupling. a** Variations of substituents at the β-position of ketones. **b** Variations of substituents attached to the carbonyl of ketones. **c** Variations of aryl bromides. All couplings were conducted on a 0.50 mmol scale (unless otherwise noted), and all yields are of purified products. TBS *tert*-butyldimethylsilyl, Boc *tert*-butoxycarbonyl, Bpin pinacolato-boron.

ketone, are well tolerated in the current system (products **3, 42–55**). Unfortunately, the reaction cannot be carried out with *o*-substituted aryl bromides (*o*-F and *o*-Me) due to the increased steric hindrance. Aryl rings can also be replaced by heteroaryls, including quinoline, benzofuran, and benzothiophene (products **56–58**). In reactions involving aryl bromides with one or more stereocenters, the catalyst determines the stereochemistry outcome instead of the substrate (products **59–62**). A reaction on a gram scale (1.22 g of product) yields coupling product **46** with similar yield and enantiomeric excess as observed in a reaction performed on a 0.50 mmol scale. The absolute configuration of products was unambiguously determined through X-ray diffraction analysis of compounds **21, 28**, and **39**.

## Applications and mechanistic observations

To illustrate the synthetic utility of this method, we have transformed the products into a variety of other useful enantioenriched compounds (Fig. 3a). For example, β-aryl ketone can be directly transformed in good yields without racemization into terminal olefin, secondary alcohol, aromatic compound, ester, and amide (products **63–67**).

Next, we applied our catalytic asymmetric synthesis of β-aryl ketone to a variety of target molecules, starting from commercially available ketones (Fig. 3b). For example, compound **68**, a potent FXR (farnesoid X receptor) agonist analog[121], can be prepared in two steps from acetophenone, via an aldol condensation followed by deoxygenative reductive cross-coupling. Pseudosinin I (**69**), a

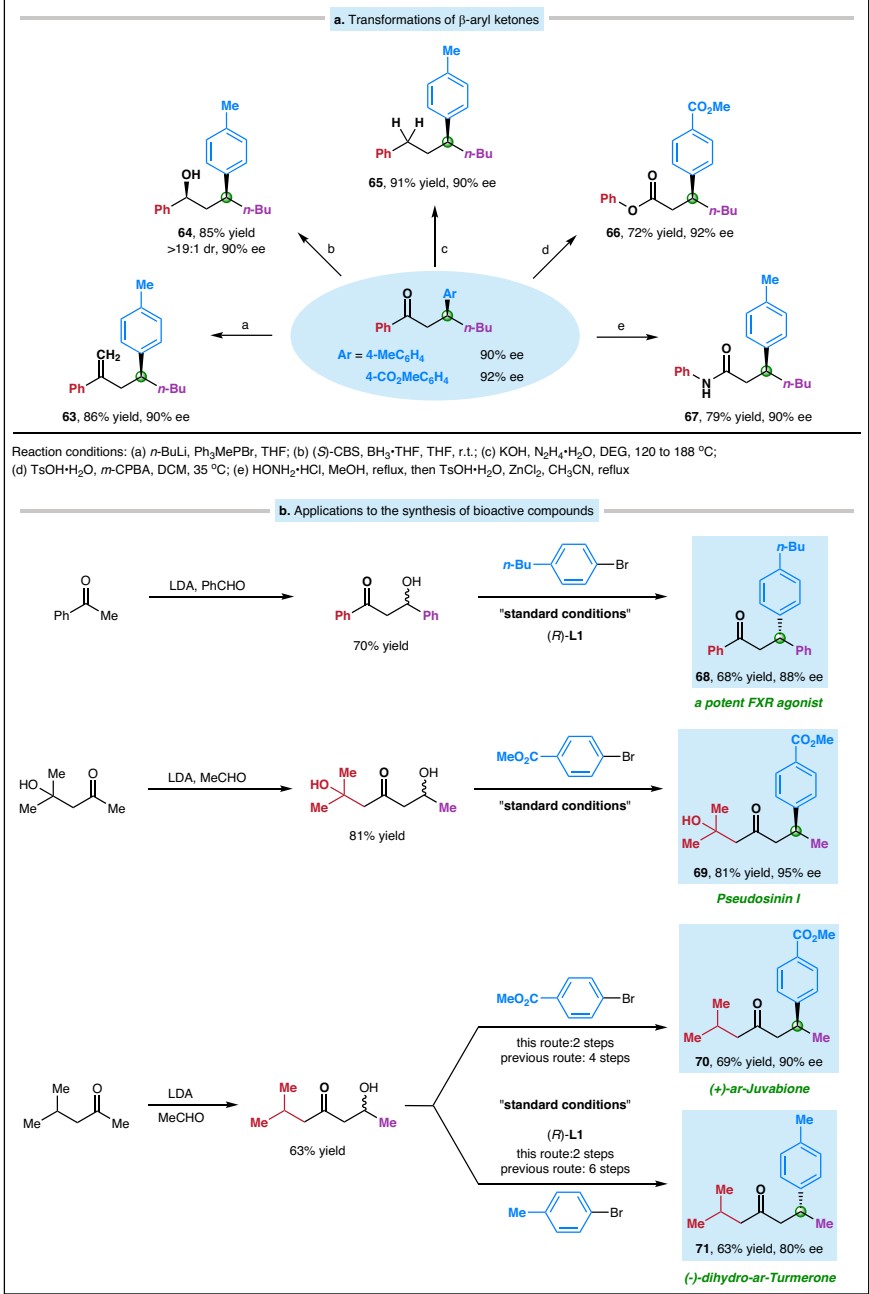

**Fig. 3 | Applications. a** Transformations into other useful families of enantioenriched compounds. **b** Applications to the synthesis of natural products and bioactive molecules. CBS Corey-Bakshi-Shibata reagent; DEG diethylene glycol, TsOH *p*-toluenesulfonic acid, *m*-CPBA *m*-chloroperoxybenzoic acid, LDA lithium diisopropylamide.

sesquiterpenoid obtained from Pseudotsuga sinensis[122], can be produced in two steps from diacetone alcohol. (+)-*ar*-Juvabione (**70**), generated earlier in four steps via an enantioselective Heck arylation, exhibits juvenile hormone properties[123]. Using our method, we can obtain β-aryl ketone **70** in two steps and 90% ee from commercially available building blocks. Another natural product, (-)-dihydro-*ar*-Turmerone (**71**), which was previously generated in six steps via an

asymmetric Michael addition and Dauben oxidation process, can be synthesized in two steps via our approach[124].

We have conducted preliminary mechanistic studies of this deoxygenative reductive cross-coupling. In 2022, Zhou and co-workers presented an elegant method for the enantioselective reductive arylation of α,β-unsaturated ketones using nickel catalyst, which provided a highly efficient approach to β-aryl ketones[125]. Mechanistic

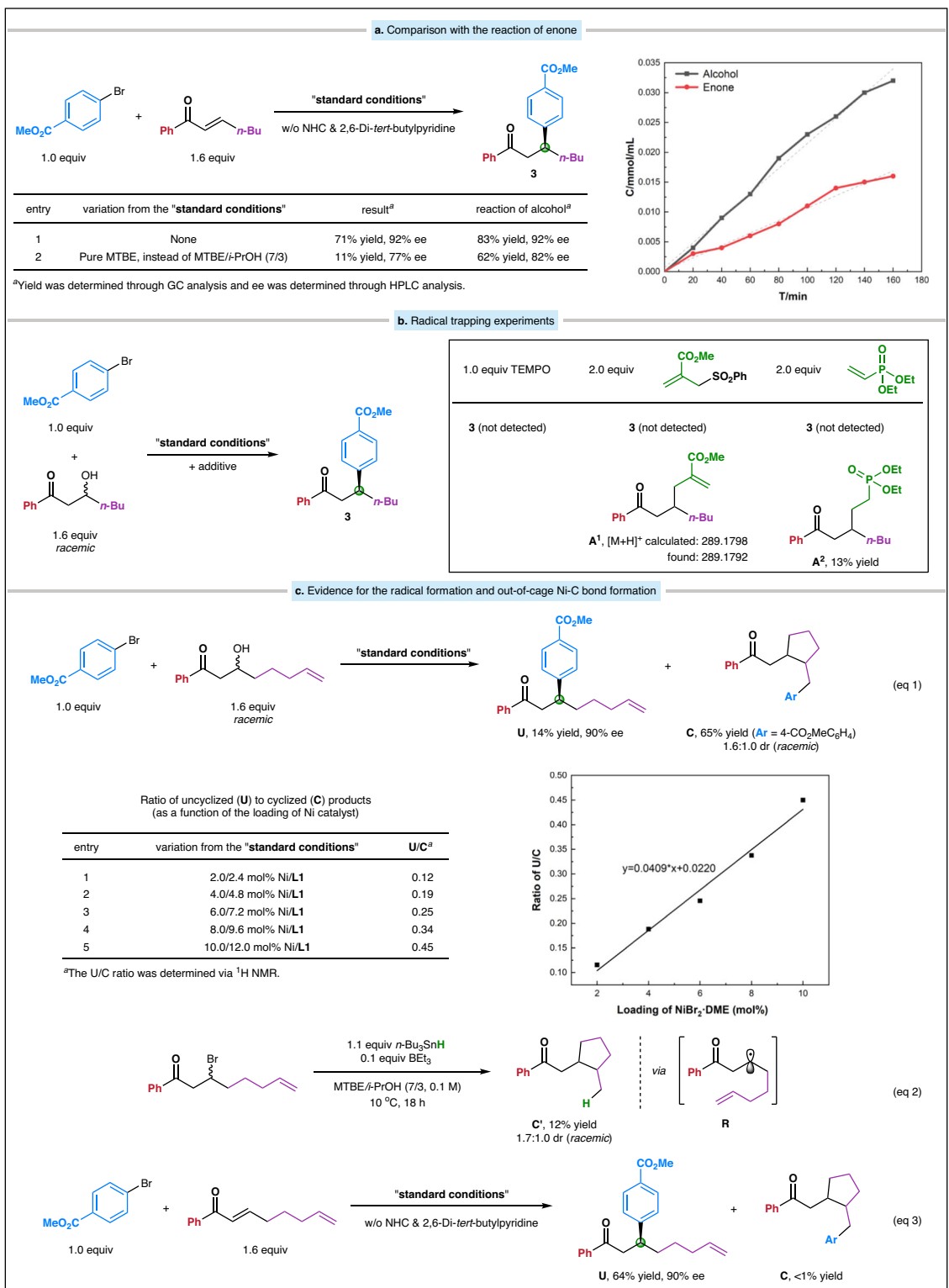

**Fig. 4 | Mechanistic studies. a** Comparison of the model reaction with the reaction of enone. **b** Radical trapping experiments. **c** Further evidence for the radical process and out-of-cage nickel-carbon bond formation. TEMPO (2,2,6,6-tetramethylpiperidin-1-yl)oxyl.

studies revealed that arylnickel(I) species inserted into enones through 1,4-addition. On the other hand, MacMillan's research have proposed the formation of organic radicals derived from alkyl alcohols[114]. In light of these findings, we are curious about whether our reaction occurs through an enone intermediate or an organic radical species.

The reaction of (E)-1-phenylhept-2-en-1-one was examined as the substrate in combination with an aryl bromide, in the absence of both NHC and 2,6-di-tert-butylpyridine. Interestingly, the desired product was successfully obtained, albeit with a slightly lower yield but maintaining the same ee (71% yield, 92% ee; Fig. 4a, entry 1). The introduction of a protic solvent ($H_2O$) was found to be crucial in Zhou's reaction. When we switched the mixed solvent system to pure MTBE, the yield for the reaction of enone was much lower compared to that of alcohol (Fig. 4a, entry 2). This observation indicates the involvement of distinct mechanisms for these two reactions. GC analysis of the model reaction (Table 1) showed that the enone species was generated in 0.10 equivalent after the initial step of NHC-alcohol adduct preparation, and maintained ~0.10 equiv throughout the entire coupling process. Kinetic studies suggested that the reaction starting from the alcohol substrate was approximately twice as fast as that starting from the enone substrate (Fig. 4a, right graph). Considering the additional time required for the formation of enone from the NHC-alcohol adduct, the likelihood of the coupling reaction proceeding through the enone intermediate is diminished.

An alternative way to differentiate between the two pathways is by observing whether an organic radical is produced. Our findings provide evidence in support of the direct deoxygenative radical pathway. For example, if 1 equivalent of TEMPO (2,2,6,6-tetramethylpiperidin-1-yl)oxyl is added to a coupling in progress, carbon–carbon bond formation essentially ceases. The addition of 2 equivalents of allylic sulfone or vinyl phosphonate to the model reaction leads to a racemic adduct $A^1$ or $A^2$ without the formation of β-aryl ketone product (Fig. 4b).

When the β-hydroxy ketone illustrated in eq 1 (Fig. 4c) was subjected to the standard coupling conditions, the uncyclized product U was generated with 14% yield and 90% ee, along with the formation of cyclized product C in 65% yield and 1.6:1.0 dr (both diastereomers are racemic). This dr value is essentially identical to that obtained in an n-Bu$_3$SnH-mediated reductive cyclization of the corresponding β-bromo ketone (1.7:1.0; Fig. 4c, eq 2), which is consistent with organic free radical R serving as a common intermediate in both processes. It has been reported that 5-hexenyl radicals cyclize with a rate constant of ~$10^5$ s$^{-1}$ [126], while the rate constant for diffusion is typically greater than $10^8$ s$^{-1}$ [127]. Thus, the identification of cyclized product C in eq 1 implies that the organic radical persists long enough to leave the solvent cage. An increase in the U/C ratio was observed with increasing nickel catalyst concentration (Fig. 4c, graph below eq 1), which suggests out-of-cage radical coupling instead of in-cage radical coupling. In contrast, the corresponding α,β-unsaturated ketone exclusively furnishes uncyclized product U (64% yield, 90% ee; Fig. 4c, eq 3), which supports a distinct 1,4-addition pathway in the coupling reaction of enone. Taken together, these observations suggest that the deoxygenative reductive cross-coupling reaction predominantly proceeds via an organic radical intermediate rather than an enone intermediate (for a proposed mechanism, see Supplementary Fig. 12).

## Discussion

We have developed a nickel-catalyzed enantioconvergent deoxygenative reductive cross-coupling of unactivated alkyl alcohol (β-hydroxy ketone) and aryl bromide in the presence of an NHC activating agent. This scalable method tolerates substituents of varying sizes on the alcohol, and displays good functional-group tolerance. This approach features the utilization of two readily available coupling partners: alkyl alcohols and aryl bromides, enabling efficient and modular access to enantioenriched β-aryl ketones including a variety

of interesting target molecules. Additional efforts to apply earth-abundant metals to useful asymmetric coupling reactions are underway in our lab.

## Methods

### General procedure for enantioconvergent deoxygenative reductive cross-coupling of alkyl alcohol and aryl bromide (alkyl alcohol: aryl bromide = 1.6: 1.0)

In a nitrogen-filled glovebox, an oven-dried 4 mL vial that contained a stir bar was charged with NiBr$_2$·DME (8.0 mg, 0.025 mmol, 5.0 mol%), (S)-L1 (6.5 mg, 0.030 mmol, 6.0 mol%), and Ir[dF(CF$_3$)ppy]$_2$(dtbbpy)PF$_6$ (9.0 mg, 0.0075 mmol, 1.5 mol%). Anhydrous isopropanol (1.5 mL) was added, and the vial was capped with a PTFE septum cap. The mixture was stirred at room temperature for 30 min, leading to a laurel-green solution. In a nitrogen-filled glovebox, a separate oven-dried 4 mL vial was charged with the alkyl alcohol (0.80 mmol, 1.6 equiv), NHC (316.5 mg, 0.80 mmol, 1.6 equiv), and a stir bar. Methyl tert-butyl ether (3.5 mL) was added, and the mixture was stirred at room temperature for 5 min. Next, 2,6-bis(tert-butyl) pyridine (179.5 μL, 0.80 mmol, 1.6 equiv) was added dropwise, and the resulting solution was stirred at room temperature for another 30 min (a white solid precipitated during this time). The suspension was filtered to furnish a homogeneous solution. In a nitrogen-filled glovebox, an oven-dried 20 mL vial was charged with the aryl bromide (0.50 mmol, 1.0 equiv), quinuclidine (67 mg, 0.60 mmol, 1.2 equiv), and a stir bar. The catalyst solution and NHC-alcohol adduct solution were transferred via syringe to the 20 mL reaction vial, followed by the addition of 4-methylpyridine (75 μL, 0.75 mmol, 1.5 equiv). The vial was transferred out of the glovebox and placed in an EtOH cooling bath at 10 °C for 5 min. Then the reaction was irradiated with blue LEDs (455 nm, 30 W) and was stirred at 10 °C for 18 h. The reaction mixture was passed through a plug of silica gel, and the vial, the cap, and the silica gel were rinsed with EtOAc. The filtrate was concentrated, and the residue was purified by flash chromatography on silica gel.

## Data availability

The experimental data and the characterization data for all the compounds generated in this study are provided in the Supplementary Information. Experimental details: general information, preparation of alkyl alcohols, catalytic enantioconvergent cross-couplings, effect of reaction parameters, cross-couplings of other alkyl alcohols, comparison between the stability of alcohol and bromide, applications, mechanistic experiments, assignments of absolute configuration, NMR spectra and determination of stereoselectivity (PDF). CCDC 2281455, 2281456, 2281457 contain the supplementary crystallographic data for this paper. These data can be obtained free of charge via www.ccdc.cam.ac.uk/data_request/cif, or by emailing data_request@ccdc.cam.ac.uk, or by contacting The Cambridge Crystallographic Data Centre, 12 Union Road, Cam-bridge CB2 1EZ, UK; fax: +44 1223 336033. All other data are available from the corresponding author upon request.

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

## Acknowledgements

We thank Prof. Gregory C. Fu (Caltech), Prof. Shu-Li You (SIOC), and Prof. Chao Zheng (SIOC) for helpful suggestions and proofreading. Support has been provided by the National Natural Science Foundation of China (Grant No. 22201216), the National Key Research & Development Program of China (Grant No. 2023YFA1508600), and the Fundamental Research Funds for the Central Universities (Grant No. 22120230252, 2023-3-YB-10).

## Author contributions

Z.-P.Y. conceived and directed the project. L.-L.Z. and Y.-Z.G. discovered and developed the reaction. L.-L.Z., Y.-Z.G. and S.-H.C. performed the experiments and collected the data. H.Y., S.-J.S. and Q.P. discussed the project with Z.-P.Y. Z.-P.Y. wrote the paper with contributions from all authors. L.-L.Z. and Y.-Z.G. contributed equally.

## Competing interests

The authors declare no competing interests.
