## [Peer Review File · Nature Communications]

Ni-Catalyzed Enantioconvergent Deoxygenative Reductive Cross-Coupling of Unactivated Alkyl Alcohols and Aryl BromidesREVIEWER COMMENTS

Reviewer #1 (Remarks to the Author):

My impression on the present manuscript is, “attractive for broad readership, while seemingly not outstanding for niche people”, which eventually led me to adopt “major revision”.

This manuscript demonstrates an enantioselective reductive deoxygenative coupling between aldol and aryl halides. Inspired by the recent achievements by MacMillan and co-workers, the authors came up with expanding their chemistry to an asymmetric reaction. The enormous (maybe too much) successful examples with a variety of substitution patterns represent the reliability of this method. Because the reaction provides a prevalent substructure found in many fields of organic chemistry, its practicality would be beyond a proof-of-concept quality. Regarding the data collection, the NMR spectra and the chromatogram qualities are basically from flawless to sufficient. The manuscript is well summarized and would be suitable to some top-notch journals. The reference section seems too comprehensive, though it is up to the authors' and the journal's preference.

What still makes me hesitate to recommend this is, however, the novelty. As already noted, the original cross-coupling reaction has already been established by MacMillan's group (refs. 114-119). The choice of the ligands (Pyox and Biox, etc.) is typical in Ni-catalyzed enantioselective reactions and is not necessarily exciting (cf. ACS Catal. 2020, 8237). Overall, the present manuscript sounds like a logical amalgam of well-established concepts.

Although being left untouched, the unsuccessful examples (Entry 19 in Table 1) can potentially receive scholarly interest because it indicates that the ketone moiety assists the reaction in addition to the enantioselection. As far as I know, Ni-catalyzed enantioselective sp²-sp³ cross-electrophile coupling reactions need a trick in most cases: either benzyl or α -carbonyl halides must be used (cf. Prof. Reisman's works). However, this limitation is not observed in the authors' reaction. Considering this, I speculate that chiral oxazoline-type ligands are not necessarily suitable for capturing alkyl radicals, but the adjacent ketone moiety can assist it. In this context, the unique concept of the present work is perhaps introduction of directing-group effect into challenging cross-electrophile coupling reactions, just as discovered in the C–H activation chemistry some time ago. If yes, I encourage the authors to focus on this a bit more. The authors may prove this by examining the efficiency of chiral oxazoline-type ligands in the reaction of non-functionalized secondary alcohols (and other radical sources).

With respect to the reaction mechanism, the authors should additionally try to exclude the possible participation of enones, which can be generated from the active NHC-esters. If enones are the true intermediates, the entire story collapses.

It might be unfortunate for the authors that the effect of β -carbonyl group for enantioselection has been reported by Zhao and Shu very recently (Sci. Adv. 2023, eadg9898). However, it does not affect my decision so much — Independent and similar discoveries are often found on trending topics.

[small problems and questions]

* In the text body, Figures are indicated like 1A, 2B..., while alphabets are uncapitalized in all Figures.

- * Table 1: stoichiometric 2,6-di-*t*-Bu-pyridine is quite expensive for lab use. Can we replace it with cheaper variants such as 2,6-lutidine?
- * Entry 4 and 13, Table 1: The non-bulky pyridine additives are indispensable to achieve high yield and ee. What is their role(s)? Can bulky bases work for this purpose?
- * Entry 17, Table 1: The volume of the reaction vial should be indicated somewhere in the table to allow readers to estimate the amount of oxygen.
- * The preparation of the reaction batch is a bit cumbersome. Can we conduct the reaction in one flask by sequentially adding the reagents?
- * Table S2, S-58: Footnotes are indicated in the table, but they are not given.
- * S-59: State how the relative configuration of 64 was determined/proposed. Is this proposed from known outcomes of CBS reduction?
- * S-106: The ¹H NMR spectrum shows non-negligible dirty signals. It should be replaced with better one.

Reviewer #2 (Remarks to the Author):

This manuscript describes the enantioconvergent deoxygenative arylation of β -hydroxyketones. The utility of this method is demonstrated by the synthesis of over fifty compounds in good yield and excellent ee, as well as a gram-scale synthesis. Given the readily accessible nature of β -hydroxyketones via aldol chemistry and the scalability of this synthetic method, I believe that this work will be useful to the synthetic community.

Nevertheless, the conceptual novelty of this work is overall low. The deoxygenative activation mode (NHC activation) was previously reported by MacMillan (Nature 2021, 598, 451), and it has already been used to facilitate nickel-catalyzed deoxygenative arylation, albeit to form racemic products. In addition, the PyrOx ligand used in this manuscript is very similar to the optimal ligand reported by MacMillan for a nickel-catalyzed deoxygenative alkylation (J. Am. Chem. Soc. 2023, 145, 7736), again to form racemic products. The authors do acknowledge these precedents clearly in the manuscript, however.

I have a few questions for the authors that I think could improve the manuscript:

- 1) Is the β -ketone motif necessary to observe high ee? One could imagine the key (PyrOx)Ni(III)-(aryl)(alkyl) intermediate forming a five-membered Ni(III) metallacycle chelate with the ketone moiety prior to reductive elimination, which could impart enhanced rigidity in the transition state. Have the authors determined the ee for non-ketone bearing substrates (e.g., 2-butanol)? On this note, could other β -substituents also have this same effect (e.g., β -amino alcohols or 1,3-diols)? I would be curious to know the ee for arylation of these non-ketone containing substrates (e.g., 2-butanol, tert-butyl N-(3-hydroxybutyl)carbamate). I believe that expanding this methodology beyond β -hydroxyketones would enhance its synthetic utility.
- 2) On that topic, the authors observe a massive drop in yield from 83% yield (92% ee) to <1% yield when switching substrates from β -hydroxyketones to β -hydroxyesters. Can the authors comment on this? In addition, more examples of unsuccessful substrates could help the reader to better understand the synthetic utility of this method.
- 3) The radical trap experiments are not completely satisfying, as the trapped products are either not observed or isolated in low yield. I would like the authors to run their reaction using

3-hydroxy-1-phenyloct-7-en-1-one as a substrate (i.e., entry 14 in the scope table with an additional methylene unit between the alcohol and alkene). Similar structures have been used previously as a mechanistic probe for radical intermediates (e.g., J. Am. Chem. Soc. 2018, 140, 12056), as a rapid 1,5 cyclization can occur. Comparing cyclized to uncyclized product could also provide an estimate for the rate at which Ni traps the alkyl radical in this system.

Reviewer #3 (Remarks to the Author):

This manuscript describes a nickel/photo-cocatalyzed asymmetric reductive cross-coupling between aryl bromides and β -hydroxyl ketones, delivering a new method to prepare chiral β -aryl ketones. In general, this reaction proceeds with good efficiency and high enantiocontrol. Since the pioneering work of Resiman (J. Am. Chem. Soc. 2013, 135, 7442, cited as Ref. 30), there have been a large number of nickel-catalyzed enantioselective reductive coupling reactions between a secondary alkyl electrophile and an electrophile of C(sp²)-X type (aryl, alkenyl, or acyl halides), which operate with a similar reaction mechanism (Ref. 30-55). The limited novelty of the current incremental work lies in the use of alcohols as the electrophilic source instead of alkyl halides. However, the in situ activation of alcohols by an NHC is also well-known in the literature (Ref. 114-119). From the perspective of reaction mechanism, the current work is a combination of the known arts of two transformations and is highly predictable. From the aspect of synthesis, the use of a stoichiometric amount of pre-synthesized NHC in the authors' work undermines its advantage over the reactions using alkyl halides, considering that the latter is easily accessible in one single step from the alcohols. Moreover, the same products could be approached via asymmetric 1,4-addition of aryl electrophiles to readily available enones, in which no extra pre-synthesized activating agent is requisite (J. Am. Chem. Soc. 2022, 144, 20249, cited as Ref. 120 without discussion and comparison). Therefore, this reviewer does not recommend the publication of this manuscript in Nature Communications due to the lack of novelty in the mechanism and the significance of synthesis.

Other comments:

1. Why does the reaction using methoxy-substituted phenyl bromide proceed in significantly lower enantiocontrol?
2. The authors need to indicate how the stereochemistry of compound 64 was assigned.

Point-by-Point Response to the Reviewers' Comments

Reviewer #1

General comment: My impression on the present manuscript is, “attractive for broad readership, while seemingly not outstanding for niche people”, which eventually led me to adopt “major revision”.

This manuscript demonstrates an enantioselective reductive deoxygenative coupling between aldol and aryl halides. Inspired by the recent achievements by MacMillan and co-workers, the authors came up with expanding their chemistry to an asymmetric reaction. The enormous (maybe too much) successful examples with a variety of substitution patterns represent the reliability of this method. Because the reaction provides a prevalent substructure found in many fields of organic chemistry, its practicality would be beyond a proof-of-concept quality. Regarding the data collection, the NMR spectra and the chromatogram qualities are basically from flawless to sufficient. The manuscript is well summarized and would be suitable to some top-notch journals. The reference section seems too comprehensive, though it is up to the authors' and the journal's preference.

What still makes me hesitate to recommend this is, however, the novelty. As already noted, the original cross-coupling reaction has already been established by MacMillan's group (refs. 114-119). The choice of the ligands (Pyox and Biox, etc.) is typical in Ni-catalyzed enantioselective reactions and is not necessarily exciting (cf. ACS Catal. 2020, 8237). Overall, the present manuscript sounds like a logical amalgam of well-established concepts.

General response: We want to express our sincere gratitude to you for the constructive feedback and helpful suggestions on our paper! Your comments have been extremely valuable in guiding us towards making the necessary improvements. To our knowledge, this method reported herein would be the first enantioconvergent deoxygenative reductive cross-coupling of alkyl alcohol (especially an unactivated alkyl alcohol) and aryl bromide. We have addressed all the comments and incorporated the changes in both the revised manuscript and the supporting information.

Comment 1: Although being left untouched, the unsuccessful examples (Entry 19 in Table 1) can potentially receive scholarly interest because it indicates that the ketone moiety assists the reaction in addition to the enantioselection. As far as I know, Ni-catalyzed enantioselective sp²-sp³ cross-electrophile coupling reactions need a trick in most cases: either benzyl or α -carbonyl halides must be used (cf. Prof. Reisman's works). However, this limitation is not observed in the authors' reaction. Considering this, I speculate that chiral oxazoline-type ligands are not necessarily suitable for capturing alkyl radicals, but the adjacent ketone moiety can assist it. In this context,

the unique concept of the present work is perhaps introduction of directing-group effect into challenging cross-electrophile coupling reactions, just as discovered in the C–H activation chemistry some time ago. If yes, I encourage the authors to focus on this a bit more. The authors may prove this by examining the efficiency of chiral oxazoline-type ligands in the reaction of non-functionalized secondary alcohols (and other radical sources).

Figure S3. Couplings of Other Alkyl Alcohols

Our response: We agree! We have conducted an assessment on various substrates for cross-coupling reactions, which includes a non-functionalized alcohol (2-butanol) and functionalized alcohols (-Ph, -OBz, -NHCbz, -NHBz, -NPh₂, etc.). The evaluation was done under standard conditions or modified conditions (MTBE/DMA as the solvent system gave better yields). All these substrates provided much lower yield and/or ee. We have included these results in our revised supporting information (Page S60, **Figure S3**).

A β -hydroxy amide was chosen, and ligand and solvent screenings were further carried out. However, no significant improvement was obtained. We have included these results in our revised supporting information (Page S61, **Figure S4**).

These findings highlight the crucial role played by the ketone moiety in the coupling reaction. To emphasize this interesting point, we have added this comment to the text:

“Under these conditions, the corresponding ester, amide, and phosphonate are not suitable coupling partners (entry 19). Additionally, substrates like 2-butanol and others that have a functional group at the β -position (-Ph, -OBz, -NHCbz, -NHBz, -NPh₂, etc.), were found to provide much lower yield and/or ee,¹²⁰ highlighting the critical role of the ketone moiety as a potential directing group in the cross-coupling reaction.¹²¹”

Figure S4. Ligand and Solvent Screenings

Comment 2: With respect to the reaction mechanism, the authors should additionally try to exclude the possible participation of enones, which can be generated from the active NHC-esters. If enones are the true intermediates, the entire story collapses.

Our response: Thank you for your insightful suggestion! We have conducted preliminary mechanistic studies. Although coupling of enone can produce the desired

product, mechanistic studies showed that the deoxygenative reductive cross-coupling reaction predominantly proceeds via an organic radical intermediate rather than an enone intermediate. We have included these results in our revised text and supporting information (Page S72-S81):

Figure 4. (a) Comparison of the model reaction with the reaction of enone. (b) Radical trapping experiments. (c) Further evidence for the radical process and out-of-cage nickel-carbon bond formation.

“We have conducted preliminary mechanistic studies of this deoxygenative reductive cross-coupling. In 2022, Zhou and co-workers presented an elegant method for the enantioselective reductive arylation of α,β -unsaturated ketones using nickel catalyst, which provided a highly efficient approach to β -aryl ketones.¹²⁶ Mechanistic studies revealed that arylnickel(I) species inserted into enones through 1,4-addition. On the other hand, MacMillan’s research have proposed the formation of organic radicals derived from alkyl alcohols.¹¹⁴ In light of these findings, we are curious about whether our reaction occurs through an enone intermediate or an organic radical species.

The reaction of (*E*)-1-phenylhept-2-en-1-one was examined as the substrate in combination with an aryl bromide, in the absence of both NHC and 2,6-di-*tert*-butylpyridine. Interestingly, the desired product was successfully obtained, albeit with a slightly lower yield but maintaining the same ee (71% yield, 92% ee; **Figure 4a**, entry 1). The introduction of a protic solvent (H₂O) was found to be crucial in Zhou’s reaction. When we switched the mixed solvent system to pure MTBE, the yield for the reaction of enone was much lower compared to that of alcohol (**Figure 4a**, entry 2). This observation indicates the involvement of distinct mechanisms for these two reactions. GC analysis of the model reaction (**Table 1**) showed that the enone species was generated in 0.10 equivalent after the initial step of NHC-alcohol adduct preparation, and maintained ~0.10 equiv throughout the entire coupling process. Kinetic studies suggested that the reaction starting from the alcohol substrate was approximately twice as fast as that starting from the enone substrate (**Figure 4a**, right graph). Considering the additional time required for the formation of enone from the NHC-alcohol adduct, the likelihood of the coupling reaction proceeding through the enone intermediate is diminished.

An alternative way to differentiate between the two pathways is by observing whether an organic radical is produced. Our findings provide evidence in support of the direct deoxygenative radical pathway. For example, if 1 equivalent of TEMPO (2,2,6,6-tetramethylpiperidin-1-yl)oxyl) is added to a coupling in progress, carbon-carbon bond formation essentially ceases. The addition of 2 equivalents of allylic sulfone or vinyl phosphonate to the model reaction leads to a racemic adduct **A**¹ or **A**² without the formation of β -aryl ketone product (**Figure 4b**).

When the β -hydroxy ketone illustrated in eq 1 (**Figure 4c**) was subjected to the standard coupling conditions, the uncyclized product **U** was generated with 14% yield and 90% ee, along with the formation of cyclized product **C** in 65% yield and 1.6:1.0 dr (both diastereomers are racemic). This dr value is essentially identical to that obtained in an *n*-Bu₃SnH-mediated reductive cyclization of the corresponding β -bromo ketone (1.7:1.0; **Figure 4c**, eq 2), which is consistent with organic free radical **R** serving as a common intermediate in both processes. It has been reported that 5-hexenyl radicals cyclize with a rate constant of approximately 10⁵ s⁻¹,¹²⁷ while the rate constant for diffusion is typically greater than 10⁸ s⁻¹.¹²⁸ Therefore, the observation of cyclized product **C** for the coupling reaction illustrated in eq 1 is consistent with the conclusion

that the organic radical has a sufficient lifetime to escape the solvent cage. Furthermore, the ratio of U/C increases as the concentration of the nickel catalyst increases (**Figure 4c**, graph below eq 1), which is expected for out-of-cage radical coupling, but not for in-cage radical coupling. In contrast, the corresponding α,β -unsaturated ketone exclusively furnishes uncyclized product U (64% yield, 90% ee; **Figure 4c**, eq 3), which supports a distinct 1,4-addition pathway in the coupling reaction of enone. Taken together, these observations suggest that the deoxygenative reductive cross-coupling reaction predominantly proceeds via an organic radical intermediate rather than an enone intermediate.¹²⁹

Comment 3: It might be unfortunate for the authors that the effect of β -carbonyl group for enantioselection has been reported by Zhao and Shu very recently (Sci. Adv. 2023, eadg9898). However, it does not affect my decision so much — Independent and similar discoveries are often found on trending topics.

Our response: Thank you for your kind understanding! We have cited this paper as reference 121 in the revised text.

Comment 4: In the text body, Figures are indicated like 1A, 2B..., while alphabets are uncapitalized in all Figures.

Our response: Thank you for catching this! We have uniformly changed them to lower case letters throughout the entire text.

Comment 5: Table 1: stoichiometric 2,6-di-t-Bu-pyridine is quite expensive for lab use. Can we replace it with cheaper variants such as 2,6-lutidine?

Our response: Yes! We tested another three bases including 2,6-lutidine, 2,4,6-collidine, and 2,6-diethylpyridine, which could also provide similar outcomes. We have included these results in our revised supporting information (Page S59, **Figure S2'**).

Figure S2'. Effect of Reaction Parameters-Extended Results

Comment 6: Entry 4 and 13, Table 1: The non-bulky pyridine additives are indispensable to achieve high yield and ee. What is their role(s)? Can bulky bases work for this purpose?

Our response: We speculate that the inclusion of a pyridine additive may engage in coordination with Ni at a certain stage, thus functioning as a co-ligand to enhance both the efficiency and enantioselectivity. Various other pyridines were examined (see above **Figure S2'**). Notably, pyridines that have one or two substituents at 2,6-positions (bulky bases) gave lower ee. In addition, more than 10 mol% of pyridine is required to maintain the good efficiency and enantioselectivity as pyridine additive could compete with quinuclidine as a base. We have included these results in our revised supporting information (Page S59, **Figure S2'**).

Comment 7: Entry 17, Table 1: The volume of the reaction vial should be indicated somewhere in the table to allow readers to estimate the amount of oxygen.

Our response: Done!

17 1.0 mL air added (4 mL reaction vial) 70 89

Comment 8: The preparation of the reaction batch is a bit cumbersome. Can we conduct the reaction in one flask by sequentially adding the reagents?

Our response: Unfortunately, no. We examined two feeding sequences (see below). Nevertheless, neither approach yielded the desired product.

entry	variation from the "standard conditions"	yield (%) ^a	ee (%) ^b
1	One flask: 1. add alcohol, 2,6-di- tert -butylpyridine, NHC , and MTBE, and stir for 30 min; 2. add Ni/ L1 , and stir for another 30 min; 3. add other agents.	0	—
2	One flask: 1. add Ni/ L1 and MTBE, and stir for 30 min; 2. add alcohol, 2,6-di- tert -butylpyridine, and NHC , and stir for another 30 min; 3. add other agents.	0	—

^a Determined through GC analysis. ^b Determined through HPLC analysis.

Comment 9: Table S2, S-58: Footnotes are indicated in the table, but they are not given.

Our response: We appreciate your careful examination of the supporting information! The footnotes are placed above the ligands.

Comment 10: S-59: State how the relative configuration of 64 was determined/proposed. Is this proposed from known outcomes of CBS reduction?

Our response: Thank you for catching this missing information! The relative

configuration of **64** was assigned according to NOESY experiment. We have included these results in our revised supporting information (Page S310-S311).

Comment 11: S-106: The ^1H NMR spectrum shows non-negligible dirty signals. It should be replaced with better one.

Our response: Done (Page S118-S119)!

Reviewer #2

General comment: This manuscript describes the enantioconvergent deoxygenative arylation of β -hydroxyketones. The utility of this method is demonstrated by the synthesis of over fifty compounds in good yield and excellent ee, as well as a gram-scale synthesis. Given the readily accessible nature of β -hydroxyketones via aldol chemistry and the scalability of this synthetic method, I believe that this work will be useful to the synthetic community.

Nevertheless, the conceptual novelty of this work is overall low. The deoxygenative activation mode (NHC activation) was previously reported by MacMillan (Nature 2021, 598, 451), and it has already been used to facilitate nickel-catalyzed deoxygenative arylation, albeit to form racemic products. In addition, the PyrOx ligand used in this manuscript is very similar to the optimal ligand reported by MacMillan for a nickel-catalyzed deoxygenative alkylation (J. Am. Chem. Soc. 2023, 145, 7736), again to form racemic products. The authors do acknowledge these precedents clearly in the manuscript, however.

General response: Thank you for your affirmative feedback and constructive suggestions, as they have undoubtedly played a pivotal role in enhancing the quality of our manuscript! We have addressed all the comments and incorporated the changes in both the revised manuscript and the supporting information.

Comment 1: Is the β -ketone motif necessary to observe high ee? One could imagine the key (PyrOx)NiIII-(aryl)(alkyl) intermediate forming a five-membered NiIII metallacycle chelate with the ketone moiety prior to reductive elimination, which could impart enhanced rigidity in the transition state. Have the authors determined the ee for non-ketone bearing substrates (e.g., 2-butanol)? On this note, could other β -substituents also have this same effect (e.g., β -amino alcohols or 1,3-diols)? I would be curious to know the ee for arylation of these non-ketone containing substrates (e.g., 2-butanol, tert-butyl N-(3-hydroxybutyl)carbamate). I believe that expanding this methodology beyond β -hydroxyketones would enhance its synthetic utility.

Figure S3. Couplings of Other Alkyl Alcohols

Our response: We agree with you that the ketone moiety plays a crucial role in the coupling reaction! We have conducted an assessment on various substrates for cross-coupling reactions, which includes a non-functionalized alcohol (2-butanol) and functionalized alcohols (the ones you mentioned and other alkyl alcohols). The evaluation was done under standard conditions or modified conditions (MTBE/DMA

as the solvent system provided better yields). All these substrates provided much lower yield and/or ee. We have included these results in our revised supporting information (Page S60, **Figure S3**).

To emphasize this interesting point, we have added this comment to the text:

“Under these conditions, the corresponding ester, amide, and phosphonate are not suitable coupling partners (entry 19). Additionally, substrates like 2-butanol and others that have a functional group at the β -position (-Ph, -OBz, -NHCbz, -NHBz, -NPh₂, etc.), were found to provide much lower yield and/or ee,¹²⁰ highlighting the critical role of the ketone moiety as a potential directing group in the cross-coupling reaction.¹²¹”

Comment 2: On that topic, the authors observe a massive drop in yield from 83% yield (92% ee) to <1% yield when switching substrates from β -hydroxyketones to β -hydroxyesters. Can the authors comment on this? In addition, more examples of unsuccessful substrates could help the reader to better understand the synthetic utility of this method.

Our response: In addition to the potential influence of directing groups, the outcomes of the reaction are also influenced by the choice of solvent and ligand. For example, when we switched the solvent system to MTBE/DMA, better yields were observed (32% yield for the β -hydroxyester; see above **Figure S3**). Furthermore, a β -hydroxy amide was chosen, and additional ligand and solvent screenings were carried out. However, no significant improvement was obtained. We have included these results in our revised supporting information (Page S61, **Figure S4**).

Figure S4. Ligand and Solvent Screenings

Comment 3: The radical trap experiments are not completely satisfying, as the trapped products are either not observed or isolated in low yield. I would like the authors to run their reaction using 3-hydroxy-1-phenyloct-7-en-1-one as a substrate (i.e., entry 14 in the scope table with an additional methylene unit between the alcohol and alkene). Similar structures have been used previously as a mechanistic probe for radical intermediates (e.g., *J. Am. Chem. Soc.* 2018, 140, 12056), as a rapid 1,5 cyclization can occur. Comparing cyclized to uncyclized product could also provide an estimate for the rate at which Ni traps the alkyl radical in this system.

Our response: Thank you for your perceptive suggestion! We examined the reaction of 3-hydroxy-1-phenyloct-7-en-1-one, which resulted in the observation of both cyclized and uncyclized products. Notably, the dr value obtained from this reaction closely resembles that obtained from the *n*-BuSnH-mediated reductive cyclization of the corresponding β -bromo ketone, implying the presence of an organic free radical as

a common intermediate in both processes. It has been reported that 5-hexenyl radicals cyclize with a rate constant of approximately 10^5 s^{-1} . Hence, it is plausible that Ni intercepts the alkyl radical at a similar rate range. In addition, the rate constant for diffusion is typically greater than 10^8 s^{-1} . Hence, the observation of cyclized product is consistent with the conclusion that the organic radical has a sufficient lifetime to escape the solvent cage (out-of-cage radical coupling, further supported by U/C ratio experiments, see below). We have included these results in our revised text and supporting information (Page S76-S80):

“When the β -hydroxy ketone illustrated in eq 1 (**Figure 4c**) was subjected to the standard coupling conditions, the uncyclized product **U** was generated with 14% yield and 90% ee, along with the formation of cyclized product **C** in 65% yield and 1.6:1.0 dr (both diastereomers are racemic). This dr value is essentially identical to that obtained in an $n\text{-Bu}_3\text{SnH}$ -mediated reductive cyclization of the corresponding β -bromo ketone (1.7:1.0; **Figure 4c**, eq 2), which is consistent with organic free radical **R** serving as a common intermediate in both processes. It has been reported that 5-hexenyl radicals cyclize with a rate constant of approximately 10^5 s^{-1} ,¹²⁷ while the rate constant for diffusion is typically greater than 10^8 s^{-1} .¹²⁸ Therefore, the observation of cyclized product **C** for the coupling reaction illustrated in eq 1 is consistent with the conclusion that the organic radical has a sufficient lifetime to escape the solvent cage. Furthermore, the ratio of U/C increases as the concentration of the nickel catalyst increases (**Figure 4c**, graph below eq 1), which is expected for out-of-cage radical coupling, but not for

in-cage radical coupling. In contrast, the corresponding α,β -unsaturated ketone exclusively furnishes uncyclized product **U** (64% yield, 90% ee; **Figure 4c**, eq 3), which supports a distinct 1,4-addition pathway in the coupling reaction of enone. Taken together, these observations suggest that the deoxygenative reductive cross-coupling reaction predominantly proceeds via an organic radical intermediate rather than an enone intermediate.¹²⁹

Reviewer #3

General comment: This manuscript describes a nickel/photo-cocatalyzed asymmetric reductive cross-coupling between aryl bromides and β -hydroxyl ketones, delivering a new method to prepare chiral β -aryl ketones. In general, this reaction proceeds with good efficiency and high enantiocontrol. Since the pioneering work of Resiman (J. Am. Chem. Soc. 2013, 135, 7442, cited as Ref. 30), there have been a large number of nickel-catalyzed enantioselective reductive coupling reactions between a secondary alkyl electrophile and an electrophile of C(sp²)-X type (aryl, alkenyl, or acyl halides), which operate with a similar reaction mechanism (Ref. 30-55). The limited novelty of the current incremental work lies in the use of alcohols as the electrophilic source instead of alkyl halides. However, the in situ activation of alcohols by an NHC is also well-known in the literature (Ref. 114-119). From the perspective of reaction mechanism, the current work is a combination of the known arts of two transformations and is highly predictable. From the aspect of synthesis, the use of a stoichiometric amount of pre-synthesized NHC in the authors' work undermines its advantage over the reactions using alkyl halides, considering that the latter is easily accessible in one single step from the alcohols. Moreover, the same products could be approached via asymmetric 1,4-addition of aryl electrophiles to readily available enones, in which no extra pre-synthesized activating agent is requisite (J. Am. Chem. Soc. 2022, 144, 20249, cited as Ref. 120 without discussion and comparison). Therefore, this reviewer does not recommend the publication of this manuscript in Nature Communications due to the lack of novelty in the mechanism and the significance of synthesis.

General response: Thank you for reviewing our paper! Despite several reports on the approaches to the direct reductive cross-coupling of alkyl alcohols and aryl halides, the asymmetric variant of this reaction remains unknown. To our knowledge, this method reported herein would be the first enantioconvergent deoxygenative reductive cross-coupling of alkyl alcohol (especially an unactivated alkyl alcohol) and aryl bromide. Consequently, we firmly believe that our findings possess considerable appeal and will captivate a diverse readership.

Comment 1: From the aspect of synthesis, the use of a stoichiometric amount of pre-synthesized NHC in the authors' work undermines its advantage over the reactions using alkyl halides, considering that the latter is easily accessible in one single step from the alcohols.

Our response: We believe our method offers an efficient approach for the coupling products! Firstly, the benzoxazolium salt used in our study can be easily prepared on a hundred-gram scale in the lab, and is already commercially available now (~10g/\$74). Secondly, our reaction does not require any additional purification of the NHC-alcohol adduct (thereby streamlining the synthesis process through a reduction in the number of steps required), while the preparation of alkyl halides usually necessitates additional and cumbersome purification. Thirdly, in certain cases, alkyl alcohol exhibits greater stability compared to alkyl halide, thus rendering alkyl alcohol a more promising starting material. For example, β -bromo ketone decomposes more easily than β -alcohol ketone (see below). We have included these results in our revised supporting information (Page S62-S63).

Figure S5. Rate of Deterioration of Alkyl Alcohol and Alkyl Bromide in Air

Figure S6. Rate of Deterioration of Alkyl Alcohol and Alkyl Bromide in $CDCl_3$

Comment 2: J. Am. Chem. Soc. 2022, 144, 20249, cited as Ref. 120 without discussion and comparison.

Our response: Thank you for your comment! We have added this comment to the text:

“In 2022, Zhou and co-workers presented an elegant method for the enantioselective reductive arylation of α,β -unsaturated ketones using nickel catalyst, which provided a highly efficient approach to β -aryl ketones.¹²⁶ Mechanistic studies revealed that arylnickel(I) species inserted into enones through 1,4-addition.”

Comment 3: Why does the reaction using methoxy-substituted phenyl bromide proceed in significantly lower enantiocontrol?

Our response: We think that the electronic effect has a subtle impact on the transition state of the reaction. Moreover, the choice of solvent system significantly impacts the outcome. In order to improve the enantiocontrol, we tested other solvents for this substrate (see below), and discovered that using PhCF₃ resulted in a better enantioselectivity of 87% ee. We have updated the results for **Figure 2** in the text accordingly.

entry	variation from the "standard conditions"	yield (%) ^a	ee (%) ^b
1	CF ₃ Ph instead of i -PrOH	62	87
2	DMA instead of i -PrOH	67	71
3	CH ₃ CN instead of i -PrOH	25	75
4	EA instead of i -PrOH	46	82
5	Dioxane instead of i -PrOH	0	-

^a Determined through GC analysis. ^b Determined through HPLC analysis.

Comment 4: The authors need to indicate how the stereochemistry of compound 64 was assigned.

Our response: Thank you for catching this missing information! The relative configuration of 64 was assigned according to NOESY experiment. We have included these results in our revised supporting information (Page S310-S311).

REVIEWER COMMENTS

Reviewer #1 (Remarks to the Author):

I am positive about publishing this after appropriate revision.

This revision includes mainly a couple of news. First, the directing effect of the ketone moiety has been disclosed. Second, the additional mechanistic study well explained the proposed radical-based mechanism. It is impressive that the enone problem was neatly dealt with by applying the other reviewer's suggestion.

I suppose the authors miss out Reviewer #2's strong encouragement to introduce JACS, 2023, 7736 in the text body. Now that the directing group effect has made the authors' reaction distinctive, I believe comparing these two achievements is rather effective to show the chemistry inside.

The least square equations in Figure 4a should be removed because the kinetics of this reaction is undetermined (I do not think this reaction is zero-order to the substrate). Likewise, it would be good to add a 1s equation to the chart in Figure 4c because the ratio is expected to proportional to the catalyst concentration.

Regarding the SI, please add comments for the newly added Figures. Motivation for these experiments and the interpretation should be important. The solvent screening for 4-bromoanisole (provided for Reviewer #3) is worth mentioning in the SI.

Reviewer #2 (Remarks to the Author):

I have examined the revised manuscript and am impressed with the detailed additional experiments performed by the authors. The points raised in my initial review have been addressed, and I recommend this for publication.

Reviewer #3 (Remarks to the Author):

The authors have revised the manuscript based on the comments of the reviewers. I insist on my previous recommendation for the following reasons:

1) nickel-catalyzed enantioselective reductive coupling reactions between a secondary alkyl electrophile (even the β -halide carbonyl compounds, cited as Ref. 121) and another carbon-electrophile have been reported many times (Ref. 30-55). Moreover, in situ activation of alcohols by an NHC is also well-known in the literature (Ref. 114-119). The novelty of this work from the mechanistic aspect is low.

2) The authors argue that the NHC they used is commercially available. However, the extra step for synthesizing this NHC by the commercial vendor should also count to evaluate the authors' reaction. Furthermore, the prices of this NHC and PBr₃ or SOCl₂, usually used for the preparation of alkyl halides, are not comparable at all. If the authors could reduce the amount of the NHC to a catalytic amount, I would consider it to be a true advance in the activation of alcohols. The authors' method as well as the MacMillan's previous work do not address the challenge but only go around it with even higher expense.

3) the authors also claim that the alcohols they used are more stable than the halides, which

is true. However, the same products could be approached starting from the much more easily accessible enones under both the known (Ref. 126) and the authors' conditions. Apparently, the authors' method requires onerous preparation of β -hydroxyl ketones and the use of a stoichiometric amount of presynthesized NHC and is thus inferior to Zhou's protocol.

Point-by-Point Response to the Reviewers' Comments

Reviewer #1

General comment: I am positive about publishing this after appropriate revision.

This revision includes mainly a couple of news. First, the directing effect of the ketone moiety has been disclosed. Second, the additional mechanistic study well explained the proposed radical-based mechanism. It is impressive that the enone problem was neatly dealt with by applying the other reviewer's suggestion.

General response: We express our gratitude once again for the positive feedback and valuable suggestions provided! We have addressed all the comments and incorporated the changes in both the revised manuscript and the supporting information.

Comment 1: I suppose the authors miss out Reviewer #2's strong encouragement to introduce JACS, 2023, 7736 in the text body. Now that the directing group effect has made the authors' reaction distinctive, I believe comparing these two achievements is rather effective to show the chemistry inside.

Our response: We agree! We have added this comment to the text:

"It is also noteworthy that the MacMillan group employed a similar pyridyloxazoline ligand to successfully couple alkyl alcohols and alkyl bromides in an achiral/racemic manner, which offers further support to the notion that the ketone moiety used in this study plays a significant role in enantiocontrol.¹¹⁸"

Comment 2: The least square equations in Figure 4a should be removed because the kinetics of this reaction is undetermined (I do not think this reaction is zero-order to the substrate). Likewise, it would be good to add a 1s equation to the chart in Figure 4c because the ratio is expected to be proportional to the catalyst concentration.

Our response: We agree! **Figure 4** has been revised accordingly.

Comment 3: Regarding the SI, please add comments for the newly added Figures. Motivation for these experiments and the interpretation should be important. The solvent screening for 4-bromoanisole (provided for Reviewer #3) is worth mentioning in the SI.

Our response: We have added a discussion section below each figure (Pages S59-S62, S64). The solvent screenings for the coupling of 4-bromoanisole have also been included (Page S60).

Reviewer #2

General comment: I have examined the revised manuscript and am impressed with the detailed additional experiments performed by the authors. The points raised in my initial review have been addressed, and I recommend this for publication.

General response: Thank you for acknowledging our efforts and providing a positive suggestion! We appreciate your encouragement!

Reviewer #3

General comment: The authors have revised the manuscript based on the comments of the reviewers. I insist on my previous recommendation for the following reasons.

General response: We sincerely appreciate your valuable time dedicated to reviewing our paper!

Comment 1: 1) nickel-catalyzed enantioselective reductive coupling reactions between a secondary alkyl electrophile (even the β -halide carbonyl compounds, cited as Ref. 121) and another carbon-electrophile have been reported many times (Ref. 30-55). Moreover, in situ activation of alcohols by an NHC is also well-known in the literature (Ref. 114-119). The novelty of this work from the mechanistic aspect is low.

Our response: We disagree! Alkyl alcohols, which exhibit significant structural diversity, are widely present in nature and easily accessible. Therefore, the direct coupling of alkyl alcohols presents a straightforward method for synthesizing complex target molecules from readily available starting materials. Despite several reports on the approaches to the direct reductive cross-coupling of alkyl alcohols and aryl halides, the asymmetric variant of this reaction remains unknown. Thus, the reaction we developed serves as a successful proof of concept. The realization of the present reaction provides valuable insights for the future development of other asymmetric deoxygenative reductive cross-couplings. Moreover, it will encourage further exploration to unlock a plethora of previously challenging asymmetric transformations. For example, quaternary stereocenters can be potentially constructed from easily accessible tertiary alcohol; however, the preparation of the corresponding tertiary alkyl halide may not be as easy or stable. Consequently, we firmly believe that our findings possess considerable appeal and will captivate a diverse readership.

Comment 2: 2) The authors argue that the NHC they used is commercially available. However, the extra step for synthesizing this NHC by the commercial vendor should also count to evaluate the authors' reaction. Furthermore, the prices of this NHC and PBr₃ or SOCl₂, usually used for the preparation of alkyl halides, are not comparable at all. If the authors could reduce the amount of the NHC to a catalytic amount, I would consider it to be a true advance in the activation of alcohols. The authors' method as well as the MacMillan's previous work do not address the challenge but only go

around it with even higher expense.

Our response: We partially agree! Implementing the deoxygenative reductive cross-coupling reaction offers distinct advantages by circumventing the need for organometallic reagents and alkyl halides, which can be vulnerable to air and moisture. Furthermore, this approach obviates the need for additional purification steps, which are often tedious and time-consuming. Although the NHC we used is relatively expensive now (may be cheap in the future as the preparation is straightforward), this type of reaction remains valuable in certain scenarios. One such example is the potential acceleration of drug development, as the deoxygenative coupling process allows for the rapid acquisition of complex target molecules utilizing easily accessible starting materials. Undoubtedly, the investigation into enantioconvergent NHC-promoted deoxygenative reductive cross-coupling remains nascent, prompting us and others to persistently address the imperative issues of expanding substrate diversity and diminishing reagent costs. Indeed, your comment about reducing the quantity of NHC aligns seamlessly with our research interests. So, we extend our gratitude once again for your valuable insights!

Comment 3: 3) the authors also claim that the alcohols they used are more stable than the halides, which is true. However, the same products could be approached starting from the much more easily accessible enones under both the known (Ref. 126) and the authors' conditions. Apparently, the authors' method requires onerous preparation of β -hydroxyl ketones and the use of a stoichiometric amount of presynthesized NHC and is thus inferior to Zhou's protocol.

Our response: Our study offers an alternative and efficient approach for the desired coupling products. Compared to Zhou's method, our approach employs photoredox catalysis instead of the traditional reducing agent, manganese, which leads to a homogeneous reaction, making it more suitable for industrial production. As far as we know, photocatalysis has demonstrated competitiveness in industrial production.